



# Insights into the abnormal increase of ozone during COVID-19 in a typical urban city of China

**Kun Zhang** [a, b#]**, Zhiqiang Liu** [a, c#]**, Xiaojuan Zhang** [a, c]**, Qing Li** [a, b]**, Andrew Jensen** [d, e]**, Wen Tan** [f]**, Ling Huang** [a, b]**, Yangjun Wang** [a, b]**, Joost de Gouw** [d, e]**, Li Li** [a, b*]

[a] School of Environmental and Chemical Engineering, Shanghai University, Shanghai, 200444, China

[b] Key Laboratory of Organic Compound Pollution Control Engineering, Shanghai University, Shanghai, 200444, China

[c] Changzhou Institute of Environmental Science, Changzhou, Jiangsu, 213022, China

[d] Cooperative Institute for Research in Environmental Sciences, University of Colorado, Boulder, Colorado, 80309, USA

[e] Department of Chemistry, University of Colorado, Boulder, Colorado, 80309, USA

[f] Tofwerk AG, Thun, Switzerland

[#] These authors contribute equally to this work.

*Correspondence:* Li Li (lily@shu.edu.cn)

## Abstract

The outbreak of COVID-19 promoted strict restrictions to human activities in China, which led to dramatic decrease in most air pollutant concentrations (e.g., $PM_{2.5}$, $PM_{10}$, $NO_x$, $SO_2$, and CO). However, abnormal increase of ozone ($O_3$) concentrations was found during the lockdown period in most urban areas of China. In this study, we conducted a field measurement targeting ozone and its key precursors by utilizing a novel proton transfer reaction time-of-flight mass spectrometer (PTR-TOF-MS) in Changzhou, which is representative for the Yangtze River Delta (YRD) city cluster of China. We further applied the integrated methodology including machine learning, observation-based model (OBM), and sensitivity analysis to get insights into the reasons causing the abnormal increase of ozone. Major



findings include: (1) By deweathered calculation, we found changes in precursor emissions contributed
5.1 ppbv to the observed $O_3$ during the Full-lockdown period, while meteorological conditions only
contributed 0.5 ppbv to the $O_3$ changes. (2) By using an OBM model, we found that although significant
reduction of $O_3$ precursors was observed during Full-lockdown period, the photochemical formation of
$O_3$ was stronger than that during the Pre-lockdown period. (3) The $NO_x$/VOCs ratio dropped
dramatically from 1.84 during Pre-lockdown to 0.79 in Full-lockdown period, which switched $O_3$
formation from VOCs-limited regime to the conjunction of $NO_x$- and VOC-limited regime. Additionally,
the decrease in $NO_x$/VOCs ratio during Full-lockdown period was supposed to increase the Mean$O_3$ by
2.4 ppbv. Results of this study investigate insights into the relationship between $O_3$ and its precursors in
urban area, demonstrating reasons causing the abnormal increase of $O_3$ in most urban areas of China
during the COVID-19 lock-down period. This study also underlines the necessity of controlling
anthropogenic OVOCs, alkenes, and aromatics in the sustained campaign of reducing $O_3$ pollution in
China.
**Keywords:** Ozone; VOCs; PTR-TOF-MS; COVID-19

## 1. Introduction

At the end of 2019, a tragic coronavirus (COVID-19) occurred, which has caused over 184 million

global infection and over 3.99 million deaths as of this writing (5 Jun 2021). To protect people's health,
China adopted strict measures to control the spread of this pandemic. Thirty provinces, autonomous
regions and municipalities have launched Full-lockdown response (also known as Level I response,
roughly from 24 Jan to 25 Feb 2020) as early as 24 Jan 2020 (Shen et al., 2021; Li et al., 2020; Huang
et al., 2020). With the effective control of COVID-19 in China, the emergency response level in most
provinces (except Hubei province, the hardest-hit region) gradually downgraded to Partial-lockdown
(Level II and Level III response, roughly after 25 Jan 2020) (Li et al., 2020), and work resumption
started. During Full-lockdown period, all the social events that may cause crowds (excluding
transportation and industries that maintained the basic operation of society) were severely restricted.
Affected by the pandemic, many factories were shut down, and the on-road traffic volume and



construction activities have been reduced significantly (Zheng et al., 2020). During Full-lockdown
period, dramatic decrease of air pollutants (e.g., $PM_{2.5}$, $NO_2$, BC) were found in China, especially in
urban areas (Fan et al., 2021; Gao et al., 2021; Li et al., 2020; Xu et al., 2020). Surprisingly, marginal
increases of $O_3$ were observed during the lockdown period in YRD region, and this seems to be
contradictory to the decrease of most air pollutants (Li et al., 2020). However, as suggested by previous
studies, the formation of $O_3$ is significantly influenced by $NO_x$/VOCs ratio and meteorological
conditions (temperature and relative humidity) (Zhang et al., 2020a; Zhang et al., 2020b). Therefore, it
is essential to investigate the changes of meteorological and emissions conditions to figure out reasons
causing the abnormal increase of $O_3$ during this pandemic.

Previous studies on the $O_3$ pollution in the YRD region have often focused on the more populated

metropolitan areas, such as Shanghai and Nanjing, which are considerably far away from the industrial
zones that are essentially responsible for the sources of $O_3$ precursors (Li et al., 2019; Zhang et al.,
2020b). Changzhou, located in the center of the Yangtze River Delta (YRD) region, is a typical city with
fast urbanization, heavy industrial structure, huge energy consumption, increasing vehicle stocks and
frequent air pollution. Therefore, it provides a more representative environment to fully elucidate the
mechanism underlying the $O_3$ pollution in the YRD region (Shi et al., 2020). In a companion paper
(Jensen et al., 2021), we also demonstrated that Changzhou is representative for the region by analyzing
both surface observations and satellite data. According to previous studies, the anthropogenic VOCs
emission in Changzhou was around $9\sim12.6\times10^4$ tons/year, among which industries was the dominant
source, accounting for $27\sim47\%$ of the total VOC emissions (Cheng et al., 2016; Fu et al., 2013). It is
notable that industrial sources together contributed over 80% of anthropogenic VOC emissions (Sun et
al., 2019). Apart from industrial sources, vehicle exhaust accounted for $9\%\sim14\%$ of total VOC
emissions (Sun et al., 2019). However, rare observation regarding VOCs characteristics during COVID-
19 in Changzhou has been conducted.

Highly time-resolved measurements of VOCs are generally much sparser and could not be easily

expanded during the lockdowns. This limits our understanding of how VOCs changed and how the
formation of ozone was affected. Here, we used a novel proton transfer reaction time-of-flight mass


spectrometer (PTR-TOF-MS, Tofwerk, Model Vocus Elf, CHE) to conduct online observation of VOCs
in Changzhou. The characteristics of VOCs and the variations of general air pollutants in each
emergency response period were analyzed. Additionally, ozone formation during each period was
investigated by an OBM model. Although terrifying impact has been caused by the COVID-19, it
provided a rare experiment to analyze the variations of VOCs and $NO_x$ due to changes of anthropogenic
activities in a typical city of China. Furthermore, results of this study offer theoretical support for
formulating refined ozone management policy in China.

## 2. Methodology


### 2.1 Field measurement


The field campaign was conducted from 8 Jan to 31 Mar 2020 at a sampling site located on the

rooftop of a building at Changzhou Environmental Monitoring Center (CEMC, 31.76° N, 119.96° E),
which was approximately 15 m above ground level. As a typical urban monitoring station, this site is in
the center of Changzhou city, surrounded by residential and commercial area, which is also adjacent to
the main transportation junction in Changzhou (Figure 1). According to local epidemic prevention
policies, we roughly classified the measurement periods into three stages: Pre-lockdown (8[th] January to
23[rd] January 2020), Full-lockdown (25[th] January to 24[th] February 2020), Partial-lockdown (25[th]
February to 28[th] March 2020) as defined in a study of the Yangtze River Delta (Li Li et al., 2020).

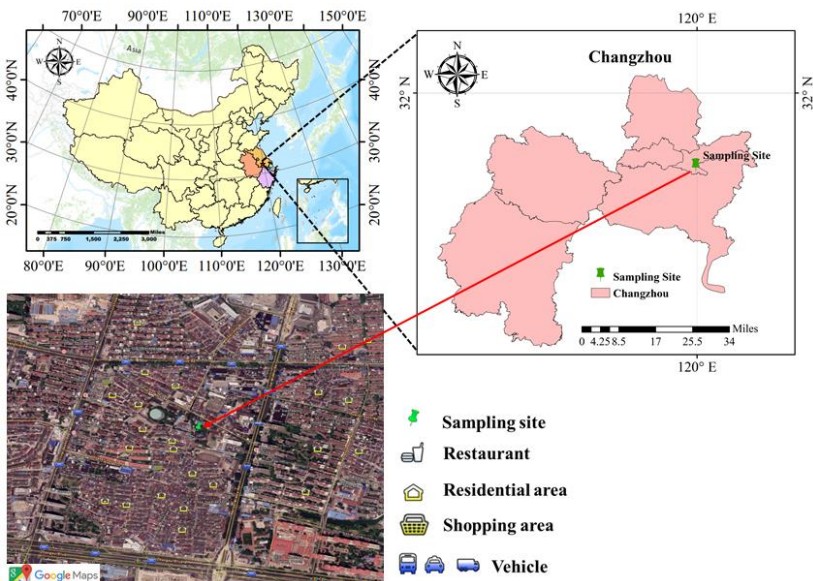


**Figure 1. Location of the sampling site in Changzhou (© Google Maps).**
From Jan 8 to Mar 27, 2020, the concentrations of traditional air pollutants ($PM_{2.5}$, $PM_{10}$, $NO_x$,
$SO_2$, CO, $O_3$) as well as meteorological parameters were monitored by a series of analyzers (Table 1).
In particular, 87 VOCs species were quantified, 59 of which were identified, by a PTR-TOF-MS with
time resolution of 1 min. Detailed measurement techniques and quality assurance and control has been
documented in detail in our companion paper (Jensen et al., 2021). Here, we just briefly introduce the
measurement. The air samples were directly inhaled into the 3 m-long tube connected to the instrument.
A priming pump, with flow rate of 4 L/min, was used to reduce the retention time of the gas sample in
the tube. To avoid blocking of inlet tube caused by particles, a particulate filter was assembled at the
front of the inlet tube. The pressure of the ion source was set as 2 mbar and the temperature of the
reaction chamber was set to 90 °C during the observation. The PTR-TOF-MS can detect most
unsaturated hydrocarbons and VOCs with functional groups but cannot detect species with proton
affinities lower than that of water, namely alkanes and small alkenes. Eighteen standard gases (including
acetonitrile, acetaldehyde, acrolein, acetone, isoprene, butanone, 2-butanone, benzene, 2-pentanone,
ethyl acetate, toluene, methyl isobutyl ketone, styrene, xylene, trimethylbenzene, naphthalene, a-pinene,
and 1,3-dichlorobenzene) with concentrations of 1 ppmv were used for the calibration of the PTR-TOF-





MS. In addition, a built-in calibration system was used to control the zero and standard gases.

**Table 1 Measurements performed during the field campaign.**

| Species/Parameter | Experimental Technique |
|---|---|
| T, RH, WS, WD and P | 2000WX, Airmax, USA |
| $O_3$ | 400E, API, USA |
| $NO_x$ (NO and $NO_2$) | T200, API, USA |
| $SO_2$ | T100, API, USA |
| CO | T300, API, USA |
| $PM_{2.5}$ | 5030, Thermo Fisher, USA |
| $PM_{10}$ | 5030, Thermo Fisher, USA |
| VOCs | Vocus Elf, Tofwerk, CHE |


## 2.2 Observation-based model

An OBM model coupled with MCM v3.3.1 was utilized to investigate the atmospheric oxidation
capability and the radical chemistry. Detailed information about the chemistry mechanism is available
on MCM website (http://mcm.leeds.ac.uk/MCM/, last access 8 Jul 2021). More than 5800 chemical
species and 17000 reactions are included in this mechanism. The photolysis frequencies (J values) were
calculated based on the trigonometric parameterization provided by MCM (Wolfe et al., 2016). Dilution
mixing within the boundary layer is considered. However, as a 0-zero model, vertical or horizonal
transport of airmasses are not involved. The observed meteorological parameters (T, RH, P), trace gases
(NO, $NO_2$, CO, $SO_2$, and VOCs) were used to constrain the model. Before each simulation, the model
was run 3 days as spin-up to reach a stable state. According to the definition of atmospheric oxidation
capability (AOC), AOC is quantified by Eq (1) (Zhu et al., 2020).

$$AOC = \sum_i k_{Y_i}[Y_i][X] \tag{1}$$

where $Y_i$ are the primary pollutants (e.g., VOCs, $CH_4$, and CO); $X$ are atmospheric oxidants (OH, $O_3$,
$NO_3$); $k_{Y_i}$ are the bimolecular rate constants for the reactions of $Y_i$ and $X$. A high value of AOC indicates
fast scavenge of primary air pollutants. Additionally, OH reactivity ($k_{OH}$), defined as the reaction rate



131 coefficients multiplied by the concentrations of the reactants with OH, is also widely used as an indicator

132 of AOC. The value of $k_{OH}$ depends on both the abundances and compositions of primary pollutants and

133 can be calculated by Eq (2).

$$k_{OH} = \sum_i k_{(OH+X_i)} \times [X_i] \tag{2}$$

134 where $k_{(OH+Xi)}$ are the reaction rate coefficients of reaction OH+$X_i$; $X_i$ are the concentrations of pollutants

135 (VOC, $NO_2$, CO, OVOC etc.) (Zhu et al., 2020).

**2.3 Trend Analysis**

137  Mann-Kendall (MK) trend test is a widely used non-parametric test method (Pathakoti et al., 2021;

138 Zhang et al., 2013), which is recommended by the World Meteorological Organization. It is applicable

139 to all distributions (that is, the data does not need to meet the assumption of normal distribution), but

140 the data should have no serial correlation. If the data has serial correlation, it will have an impact on the

141 significance level (p value). In this study, the MK trend analysis was performed for individual VOC

142 concentrations during Pre-lockdown and Full-lockdown period. Detailed description of this method

143 could be found in the study of Pathakoti et al. (2021) and Alhathloul et al. (2021). A positive z value

144 from the MK test indicates increasing trend of the target compound. On the contrary, a negative z value

145 suggests the target compound was decreasing.

146  Sen's slope, a non-parametric test proposed by Sen (1968), is also used in this study to assess the

147 rate of change in individual VOC concentrations. Sen's slope (Q) is mathematically represented by the

148 following equations.

$$Q = median(SS_{ij}) \tag{3}$$

$$SS_{ij} = \frac{x_j - x_i}{j - i}, 1 \le i \le j \le n \tag{4}$$

149 where $x_j$ and $x_i$ are concentrations of VOC specie x at time $j$ and $i$ (1≤$i$≤$j$≤$n$), respectively. $SS_{ij}$ is the

150 linear slope between time $i$ and $j$, and Q is the median of $SS_{ij}$. Positive and negative Q values indicating

151 increasing or decreasing trend of VOC specie x, respectively.

**2.4 Deweathered model**





The observed concentrations of $O_3$ could be influenced by meteorological conditions,
emissions/chemistry. To quantitatively assess the contribution of meteorological conditions and
emissions/chemistry, the deweathered $O_3$ concentrations was calculated based on the random forest (RF)
approach. The number of trees in the RF model was set as 300, the minimal node size was five, and the
number of samples was 300. Hourly data of Unix date (number of seconds since 1970-01-01), Julian
day, weekday, hour of day, wind speed (WS), wind direction (WD), temperature (T), relative humidity
(RH), and pressure (P) were used for the deweathered calculation of $O_3$. More details of this model
could be found in the study of Grange and David (2019).

## 3. Results and discussion

### 3.1 Overview of the field campaign

Figure 2 shows the meteorological conditions during the observation. During the whole experiment,
the prevailing WD was southeast. The average T and RH was $9.9 \pm 5.1°C$ and $58.9 \pm 17.1\%$, respectively.
Compared to Pre-lockdown period, the concentrations of $PM_{2.5}$, $PM_{10}$, $SO_2$, NO, $NO_2$, TVOC and CO
during Full-lockdown period decreased by 48%, 42%, 11%, 65%, 58%, 33% and 39%, respectively. It
is interesting to note that the decreasing ratio of VOC : $NO_x$ is around 1:2. Meanwhile, the average $O_3$
concentrations in Full-lockdown period was 67% higher than that during Pre-lockdown period. To
estimate whether the increase of $O_3$ during Full-lockdown period is abnormal, we summarized the
meteorological conditions and $O_3$ concentrations during the same period in 2020 and 2019 (Table 2 and
Figure 3). It should be noted that, considering the influence of Chinese New Year, the corresponding
period in 2019 was decided according to lunar calendar. Compared to Full-lockdown period in 2019,
the mean $O_3$ concentration in 2020 was 5.5 ppbv higher (Figure 2). The T and RH in Full-lockdown
period in 2020 was ~1.6 °C higher and 6.1% lower than that in the same period in 2019, while the P and
WS were comparable during the same period in 2020 and 2019.  The relatively higher T and lower RH
condition was supposed to be in favor of $O_3$ formation during the Full-lockdown period in 2020.
Additionally, we compared the hours of adverse weather conditions (with RH >70% and WS <1m/s)
and found that the meteorological condition during Full-lockdown period in 2020 restrained the dilution





of air pollutants (Table 2). Therefore, the meteorological conditions during Full-lockdown period in
2020 seemed to favor $O_3$ formation. However, changes in $O_3$ concentrations could be a result of the joint
effect of meteorological conditions and emissions/chemistry, the following sections would discuss the
influences respectively.

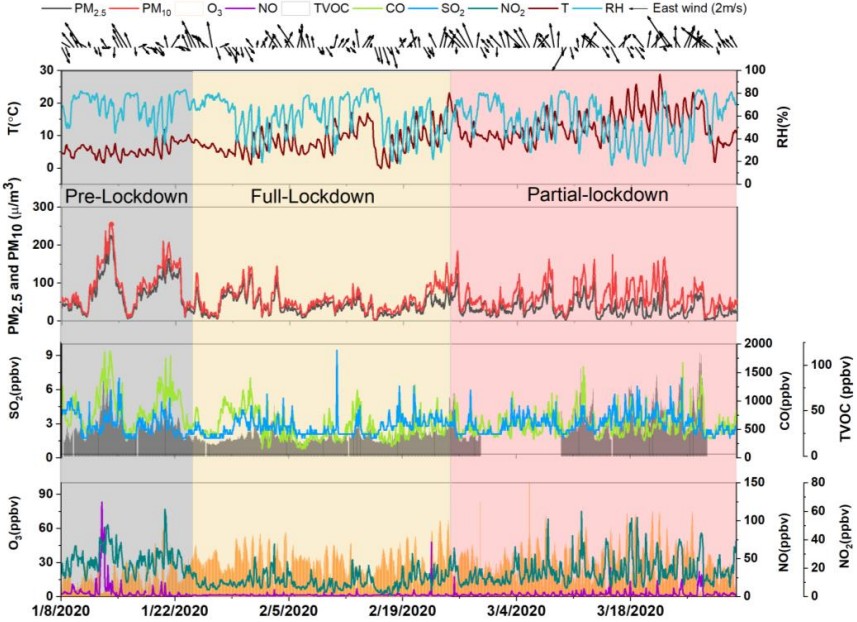


**Figure 2 Time series of meteorological parameters and air pollutants during the whole observation.**
**Table 2 Comparison of average meteorological conditions during Pre-lockdown, Full-lockdown, and Partial-**
**lockdown period in 2020 and the same period in 2019.**

| Periods | Date | P (hPa) | RH (%) | T (°C) | Precipitation (mm) | WS (m/s) | Adverse weather conditions* | |
|---|---|---|---|---|---|---|---|---|
| | | | | | | | Number of hours (h) | Proportion (%) |
| Pre-lockdown | (2020.1.8-1.24) | 1025.4 | 84.9 | 4.8 | 0.13 | 1.8 | 54 | 17.5% |
| Same period in 2019 | (2019.1.19-2.4) | 1025.6 | 72.7 | 5.2 | 0.05 | 1.9 | 70 | 18.5% |
| Full-lockdown | (2020.1.25-2.24) | 1025.6 | 73.0 | 7.3 | 0.09 | 2.1 | 103 | 15.6% |
| Same period in 2019 | (2019.2.5-3.7) | 1024.1 | 79.1 | 5.7 | 0.15 | 2.1 | 82 | 13.7% |
| Partial-lockdown | (2020.2.25-3.31) | 1018.9 | 69.5 | 12.1 | 0.11 | 2.4 | 106 | 12.3% |
| Same period in 2019 | (2019.3.8-4.12) | 1017.6 | 64. | 13.8 | 0.02 | 2. | 72 | 8.3% |

* The non-rainfall periods with RH less than 70% and WS below 1m/s were defined as adverse weather condition.

**3.2 Mechanism affecting the abnormal $O_3$ increase**

### 3.2.1 Meteorological perspective


Deweathered $O_3$ concentrations were calculated based on the model described in Section 2.4. The

random forest model grown for $O_3$ at the sampling site had $R^2$ values of 0.84, therefore, the model shows
good performance for $O_3$ predictions. The difference between observed ($O_{3,Obs}$) and weather-normalized
$O_3$ ($O_{3,Normal}$) can be regarded as the meteorological influence ($O_{3,Met}$). In addition, the difference
between $O_{3,Obs}$ concentrations in different years could be considered as the influence of emissions
($O_{3,Emi}$). Figure 3 exhibited the average $O_{3,Obs}$, $O_{3,Normal}$, $O_{3,Met}$ during the same periods in 2019 and 2020,
respectively. The mean $O_{3,Normal}$ during Pre-lockdown and Partial-lockdown periods in 2019 and 2020
were close, suggesting the similar emission condition during these periods in 2019 and 2020. During
Full-lockdown period in 2020, $O_{3,Met}$ contributed +0.5 ppbv to $O_{3,Obs}$, which is consistent with the rough
summary of meteorological conditions in Table 2, confirming that the weather conditions during the
Full-lockdown period in 2020 favored $O_3$ formation. However, the average $O_{3,Normal}$ during Full-
lockdown period in 2020 was 5.1 ppbv higher than that in 2019, indicating that improper decline of
precursor emissions was possibly the key reason for the abnormal increase of $O_3$ during Full-lockdown
period in 2020.

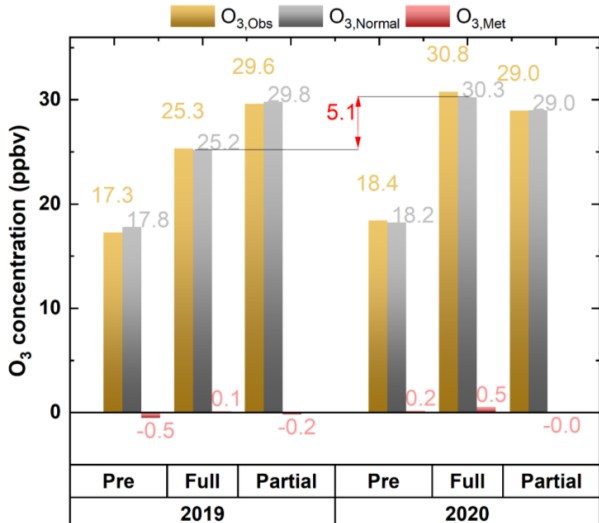

**Figure 3. Comparison of observed (Obs), weather-normalized (Normal), and meteorological-factors-infected**
**(Met) $O_3$ concentrations during the same period in 2019 and 2020.**



### 3.2.2 Ambient VOCs


As mentioned above, the changes in $O_3$ precursor emissions strongly affected the $O_{3,Obs}$, and the
changes in VOCs and $NO_x$ emissions would eventually be reflected by the observed concentrations of
individual VOCs and $NO_x$. Therefore, the concentrations of each VOC group in different periods were
summarized (Figure 4). OVOCs dominated the total VOCs (TVOC) concentrations during the whole
observation, with a daily average concentration of $21.44 \pm 10.27$ ppbv. During Full-lockdown period,
the TVOC dropped to $22.19 \pm 7.9$ ppbv, which was mainly affected by the decrease in industrial activities
and traffic volume. The most obvious drop was found in aromatics (~54%), followed by OVOCs (~27%),
alkenes (~26%), nitrogen hydrocarbon (~25%), and other VOCs (~21%). Additionally, the discrepancy
of daytime and nighttime VOCs concentrations during different periods were compared (Figure 4 (A)).
Interestingly, the concentration of each VOCs group exhibited higher values during nighttime, which
was caused by the low atmospheric oxidation condition and the low atmospheric boundary layer height
(Maji et al., 2020; Valach et al., 2015).

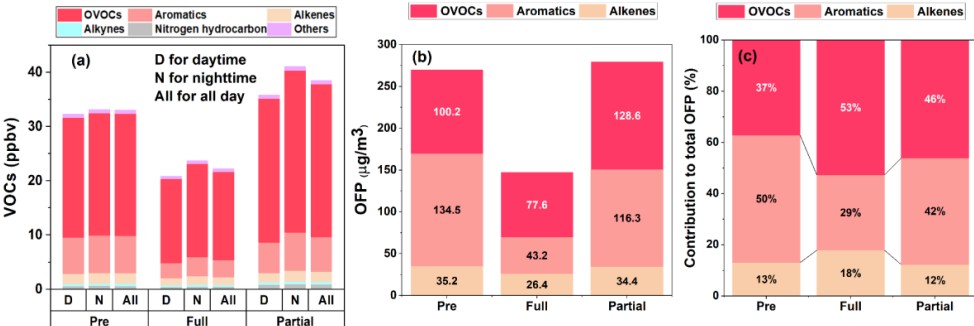

**Figure 4.   Comparison of daytime and nighttime VOCs concentrations (A), average OFP (B), and contribution**

**to total OFP (C) during different periods.**

Furthermore, the average concentrations of individual VOCs during different periods were
summarized in Figure 5. Most VOC species exhibited an 'U' shape trend during the whole observation,
except for several VOCs (such as formaldehyde (HCHO) and methanol), which showed an increasing
pattern. It should be noted that the measurement of HCHO could be strongly influenced by humidity.
Since within the drift tube, the back reaction, which converse the protonated HCHO back into HCHO,





is highly humidity dependent (Inomata et al., 2008; Warneke et al., 2011). To quantitatively evaluate the
changes of individual VOC concentrations from Pre-lockdown to Full-lockdown period, when the
variations of each VOCs are obvious, we applied MK trend test and Sen's slope analysis based on the
hourly average VOCs concentration data.

Table 3 lists the top 10 VOCs species with decreasing pattern from Pre-lockdown to Full-lockdown

period. Interestingly, toluene, benzene and xylene exhibited the most significant decreasing pattern, with
a slope of $7.73\times10^{-4}$, $7.36\times10^{-4}$, and $7.20\times10^{-4}$ ppbv h$^{-1}$, respectively. As for $NO_x$ and TVOC, the slope
was $-1.62 \times 10^{-2}$ and $5.48\ 10^{-3}$ ppb h$^{-1}$. This result corresponds with the drastic drop of industrial
activities and traffic volumes, which are key sources of aromatics and $NO_x$, from Pre-lockdown to Full-
lockdown period. Other VOCs, such as ethyl-acetate, acetic acid, acetaldehyde, diethyl sulfide, ethanol,
butanol and acrolein are also tightly associated with industrial processes, thereby showed decreasing
trend from Pre-lockdown to Full-lockdown period. Additionally, the average diurnal variations of
acetonitrile, dimethyl formamide (DMF), and styrene, which are tracers of biomass burning and
industrial emission, respectively, exhibited significant reduction during Full-lockdown period (Figure
S1), also indicating strong decrease in these emissions. However, formaldehyde and methanol exhibited
increasing trend, with a slope of $12.78\times10^{-4}$ and $6.35\times10^{-4}$ ppbv h$^{-1}$, respectively. This could be
explained by the secondary formation of HCHO and methanol, which was promoted under better
oxidation condition in Full-lockdown period.





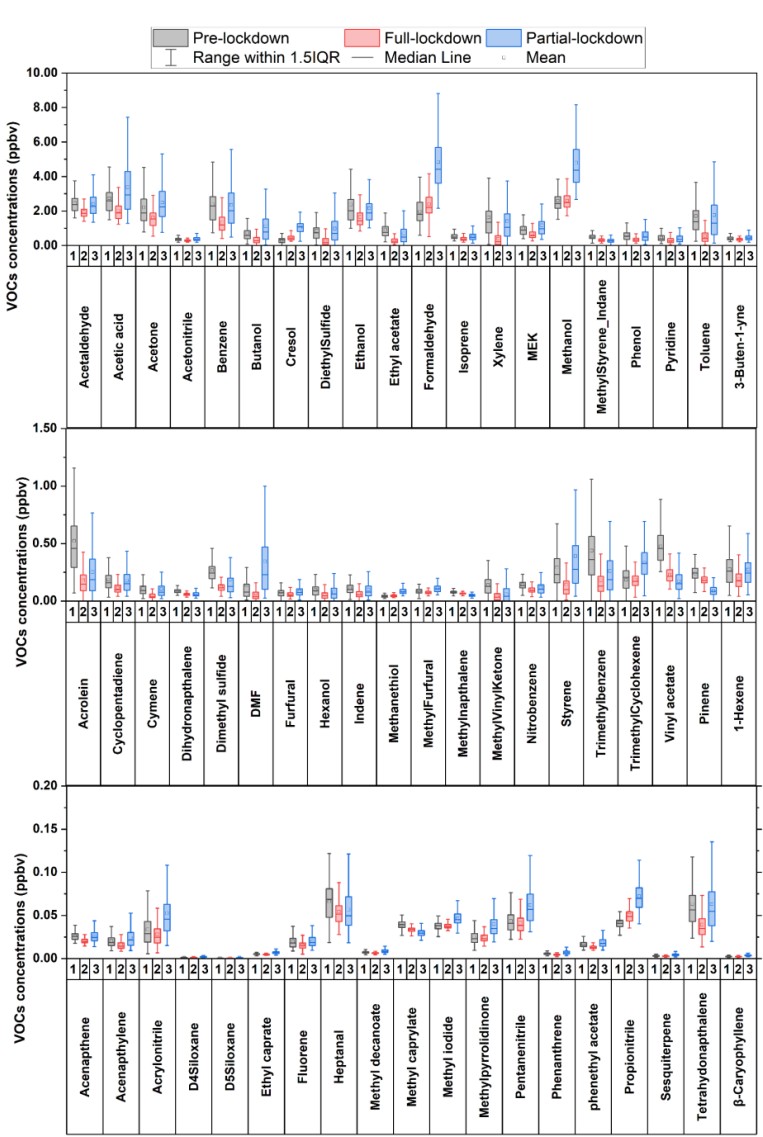


**Figure 5. Concentrations of individual VOC species during different period.**
**\*MEK, DMF, are abbreviation of Methyl ethyl ketone and dimethylformamide, respectively.**








**Table 3. Top 10 VOCs with decreasing trend from Pre-lockdown to Full-lockdown**

| VOC | Z value | Q *10000 (ppbv h$^{-1}$) | VOC | Z value | Q *10000 (ppbv h$^{-1}$) |
|---|---|---|---|---|---|
| Toluene | -14.02 | -7.73 | Acetaldehyde | -10.31 | -3.95 |
| Benzene | -9.65 | -7.36 | Diethyl sulfide | -9.15 | -3.16 |
| xylene | -12.38 | -7.20 | Ethanol | -5.48 | -3.09 |
| Ethyl-acetate | -18.53 | -5.20 | Butanol | -10.42 | -2.83 |
| Acetic acid | -6.79 | -4.12 | Acrolein | -15.48 | -2.76 |

**3.2.3 Chemistry perspective**
The reactivities of different VOCs varies significantly, hence, ozone formation potential (OFP) is
used in this study to assess the potential contribution of active VOCs (including alkenes, aromatics and
OVCOs) to $O_3$ formation on the same basis, and it can be calculated by formula (5):

$$OFP_i = MIR_i \times [VOC_i] \qquad (5)$$

where $MIR_i$ is the ozone formation potential coefficient for a given VOC specie $i$ in the maximum
increment reaction of $O_3$, acquired from Carter (2009); $[VOC_i]$ is the concentration of VOC species $i$ (in
$\mu g/m^3$). The time series of total OFP is shown in Figure 6. The average OFP in Pre-lockdown, Full-
lockdown, and Partial-lockdown period was 269.4 ± 146.0, 147.2 ± 72.4, 279.3 ± 168.6 $\mu g/m^3$,
respectively. The trend of the total OFP indicates the drastic decrease of VOCs reactivities from Pre-
lockdown to Full-lockdown period. During Pre-lockdown period, aromatics were the dominant OFP
contributor (49%), followed by OVOCs (38%) and alkenes (13%) (Figure 4). Among aromatics, xylene
exhibited the maximum OFP value (68.6 ± 59.3 $\mu g/m^3$), followed by acetaldehyde (28.8 ± 6.4 $\mu g/m^3$),
toluene (25.7 ± 20.1 $\mu g/m^3$) trimethylbenzene (25.4 ± 15.8 $\mu g/m^3$), and formaldehyde (22.7 ± 9.1 $\mu g/m^3$)
(Figure S2), suggesting that anthropogenic emissions could be the main source of secondary formation
of $O_3$ during Pre-lockdown period. Compared to Pre-lockdown period, the OFP of aromatics decreased
dramatically (-91.2 $\mu g/m^3$) during Full-lockdown period (Figure 4 (B)), which was mainly attributed to
the rapid decline of human activities (e.g., transportation and industry). However, the OFP of alkenes
and OVOCs only decreased by 8.9 and 22.5 $\mu g/m^3$, respectively. As for alkene, this could be explained
by their chemical reactivities, which led to the fast degradation after emission. As for OVOCs, the
secondary formation could compensate the decrease in primary emissions. The OFP values of aromatics





and alkenes during Pre-lockdown and Partial-lockdown period are comparable, but OVOCs exhibited
higher OFP contribution (~46%) in Partial-lockdown period, which could be attributed to the higher
AOC condition during Partial-lockdown period. To compare the average reactivity of VOCs during
different periods, we calculate the mean MIR in each period. As shown in Figure 7, the average MIR
during Pre-lockdown, Full-lockdown, and Partial-lockdown period was 3.85, 3.53 and 3.68 (g O$_3$/g
VOC), respectively. This result suggests that the VOCs in Partial-lockdown should produce less O$_3$ than
that in Pre-lockdown, and Partial-lockdown period. This is inconsistent with the observation, which
shows relatively higher O$_3$ concentration during Full-lockdown period. However, the formation of O$_3$
was sensitive to the ratio of NO$_x$/VOCs and meteorological conditions, which can be significantly
different in each period. As shown in Figure 7, the average NO$_x$/VOCs ratio in the three periods (shown
in) was 1.84, 0.79, and 0.84, respectively, suggesting more NO$_x$ was eliminated during Full-lockdown
period, which could further influence the sensitivity of O$_3$ formation.

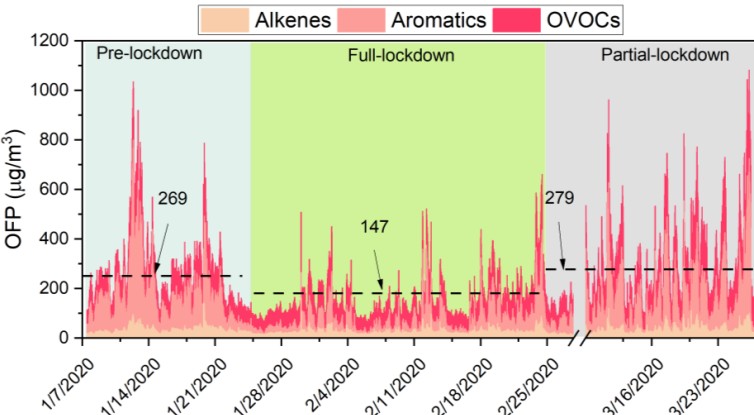

**Figure 6. Time series of OFP during the whole observation period (dash lines represent the average OFP value**
**during each period)**



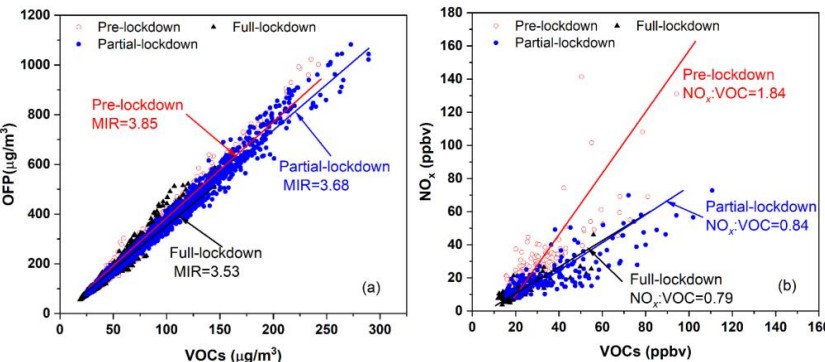

**Figure 7. Plot of average MIR and NO$_x$ vs VOCs during three periods.**
To investigate the detailed formation mechanism of O$_3$ in each period, three cases (January 19th,
February 1st, March 14th) with stagnant meteorological conditions were chosen. The index of agreement
(IOA) of O$_3$ is 0.80, indicating that the model can capture the daytime variation of O$_3$. The simulated
daytime OH concentrations exhibited an increasing trend from January 19 to March 14, with an average
value of $0.36 \pm 0.27 \times 10^6$, $0.75 \pm 0.54 \times 10^6$ and $1.18 \pm 0.78 \times 10^6$ molecules cm$^{-3}$, respectively. This
could be attributed to the increasing solar radiation and temperature from January to March. To analyze
the atmospheric oxidation, we calculated the AOC according to Eq(1). The average daytime AOC on
Jan 19th, Feb 1st, and Mar 14th was $0.26 \pm 0.35$, $0.23 \pm 0.33$, and $0.31 \pm 0.38$ molecules cm$^{-3}$ s$^{-1}$,
respectively (Figure 9). Comparatively, these values are much lower than those simulated for Shanghai
and Beijing (Liu et al., 2012; Zhu et al., 2020; Zhang et al., 2021) in summer, mainly due to the
meteorological conditions in winter season. It is notable that the simulated OH on Jan 19th was
significantly lower than that on Feb 14th, but the AOC on Jan 19th was comparable to that on Feb 1st.
This should be ascribed to the abundant primary pollutants, which efficiently react with OH, during Pre-
lockdown period.

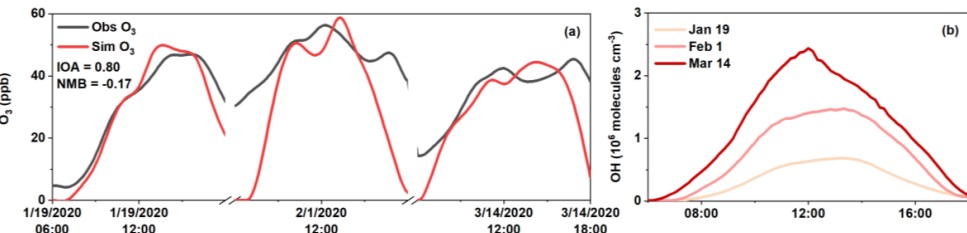

**Figure 8. Comparison of simulated and observed O₃ (a) and simulated daytime OH concentrations (b) in three cases.**

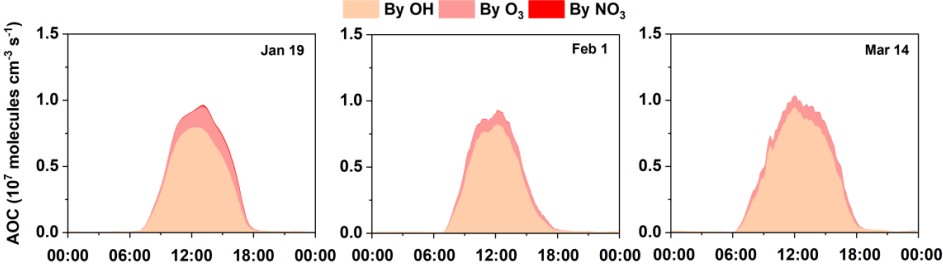

**Figure 9. Diurnal variation of AOC in three cases**

The daytime variations of OH reactivity calculated by OBM model are exhibited in Figure 10, including the contribution from measured pollutants (e.g., VOCs, NO$x$, and CO) and model-simulated species (OVOCs). Generally, the $k_{OH}$ assessed at Changzhou was in the range of 9~32 s$^{-1}$, which was comparable to that calculated for other cities in China (e.g., Shanghai (4.6~25 s$^{-1}$, (Zhu et al., 2020)), Chongqing (15~25 s$^{-1}$, (Tan et al., 2019)) and Beijing (15~25 s$^{-1}$, (Tan et al., 2019))). It is obvious that OH reactivity peaked in the morning, with maximum values of 31.76, 17.98, and 17.30 s$^{-1}$, respectively. The OH reactivity from NO₂ exhibited obvious daytime variation, especially during the morning rush hour, which lead to the peak $k_{OH}$ value during morning. The OH reactivity ($k_{OH}$) on Feb 1$^{st}$ was much lower than that in the other two cases, which was mainly due to the abundance of emissions during Pre-lockdown and Partial-lockdown period. Compared to Jan 19$^{th}$, the $k_{OH}$ from NO₂ on Feb 1$^{st}$ and Mar 14$^{th}$ showed lower levels, with an average value of 2.62 and 3.35 s$^{-1}$, respectively. This corresponds with the dramatic drop of traffic volume during lockdown periods. Similarly, compared to Jan 19, the $k_{OH}$ from alkenes and aromatics were lower on Feb 1$^{st}$ and Mar 14$^{th}$. It is notable that the OH reactivity from

OVOCs kept almost stable in the three cases, this was mainly attributed to the stable OVOC
concentrations during these cases.

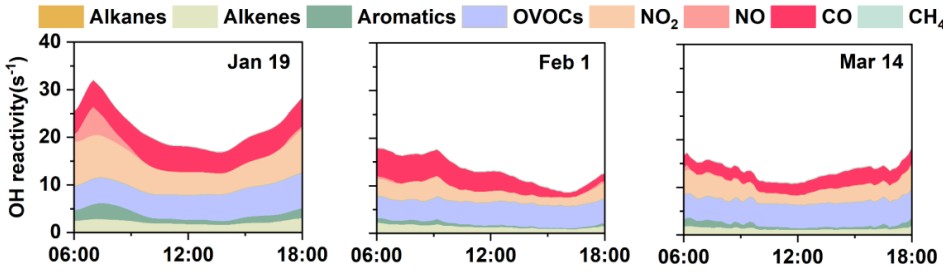

**Figure 10. Daytime variation of OH reactivity in three cases**

To investigate the variation of $O_3$ during different periods, the formation and loss pathways of $O_3$

were calculated (Figure 11). The formation of $O_3$ ($P(O_3)$) was dominated by $HO_2+NO$ and $RO_2+NO$
pathways. Although the average MIR during Full-lockdown period was the minimum among the three
periods, the $P(O_3)$ on Feb 1st was higher than that on Jan 19th. This could be attributed to the higher
AOC and better photochemical conditions during Full-lockdown period. Similarly, much higher $P(O_3)$
was found on March 14th. To avoid the influence of meteorological conditions and test the potential
mean $O_3$ (MeanO₃) concentrations under different $NO_x$/VOCs ratios, a series of scenario analyses were
performed based on the average condition during the whole observation, and the isopleths of MeanO₃
concentrations are exhibited in Figure. 12. Note that the value of temperature and photolysis frequencies
(J values) in the scenario analyses could be higher than the actual value during Pre-lockdown period
and could further lead to overestimation of simulated MeanO₃ during Per-lockdown period. Additionally,
the VOCs concentrations mentioned in this section only represent the VOC species in the MCM
mechanism. By connecting the inflection points in each $O_3$ isopleth, we get the ridge line, which divides
the whole regime into $NO_x$-sensitive and VOCs-sensitive regimes (Figure. 12). During Pre-lockdown
period, the $O_3$ formation was in VOC-limited regime (triangles in Figure. 12), with an average
$NO_x$/VOC ratio of 1.84. As for Full-lockdown period, significant decrease of $NO_x$ and VOC emissions
was observed, and the $NO_x$/VOCs ratio dropped to 0.79, which gradually switched the $O_3$ formation to
the junction of VOCs-limited and $NO_x$-limited regimes, especially on Feb 16th and Feb 17th (circles in
the red rectangle in Figure. 12), when the $O_3$ formation went into $NO_x$-limited regime. During Partial-





lockdown period, increasing of VOCs and $NO_x$ emission again dragged the formation of $O_3$ back into
VOCs-limited regime (triangles in Figure. 12). Interestingly, although a great deal of $NO_x$ and VOCs
emissions were diminished during Full-lockdown period, the average $MeanO_3$ in Full-lockdown was
supposed to be 2.4 ppbv higher than that in Pre-lockdown period. This result is consistent with the trend
of the observed MDA8 $O_3$ and the results of the deweathered calculation. Therefore, the improper
$NO_x$/VOCs reduction ratio and further influence on chemistry was the key reason for the abnormal
increase of $O_3$ during Full-lockdown period in Changzhou in 2020.

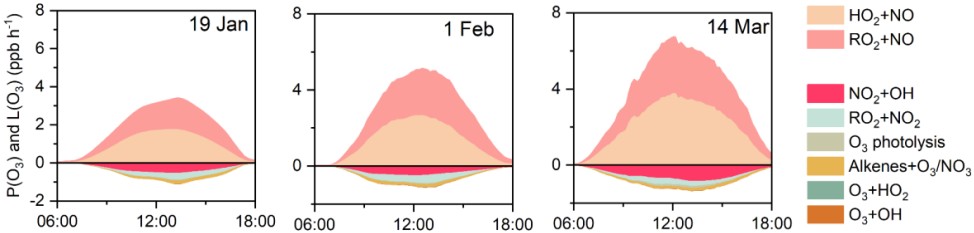


**Figure 11. Daytime variation of P($O_3$) and L($O_3$) in three cases**




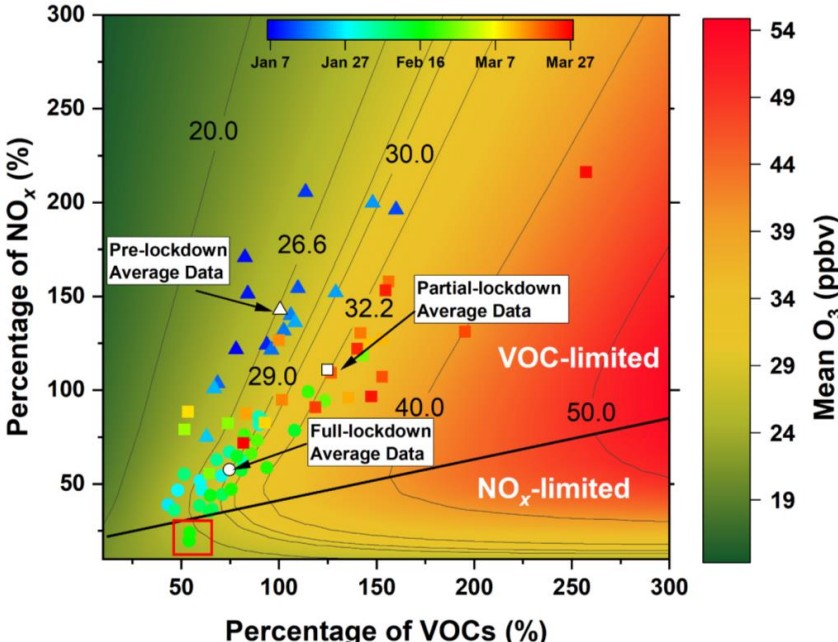


**Figure. 12 MeanO$_3$ isopleth. The colored circles, triangles, and rectangles represent the daily average**


**concentrations of NO$_x$ and VOCs during Pre lockdown, Full-lockdown, and Partial-lockdown period,**


**respectively. The white circle, triangle, and rectangle indicates the average NO$_x$ and VOCs concentrations**


**during Pre lockdown, Full-lockdown, and Partial-lockdown period, respectively.**


The scenario analyses raise a question: how much O$_3$ would change as a function of reduction of
NO$_x$ and VOCs? Therefore, the reduction percentage of O$_3$ ($\Delta$O$_3$/O$_3$) during Pre-lockdown period as a
function of reduction of VOCs and NO$_x$ were calculated, and the result could be regarded as a potential
to control O$_3$ pollution. Based on the VOCs species in MCM v3.3.1, we classified the measured VOCs
into four groups: alkenes (n-butene); aromatics (including benzene, toluene, phenol, xylene, styrene,
cresol, and trimethylbenzene); OVOCs (including methanol, ethanol, formaldehyde, aldehyde, acrolein,
methyl vinyl ketone, methyl ethyl ketone, ethyl acetate, methyl isobutyl ketone, hexanol, and heptanal);
and BVOCs (isoprene, pinene, and caryophyllene). The results in Figure 13(a) indicate that more
reduction potential of O$_3$ could be achieved by diminishing aromatics, followed by BVOCs, OVOCs,
and alkenes. It should be noted that many light alkanes and active alkenes, such as ethene and propene,




could not be measured by the PTR-TOF-MS and might further lead to the underestimation of ozone
sensitivity to alkanes and alkenes. Additionally, this comparison has a drawback of being influenced by
the concentrations of VOCs. To normalize the influence of concentrations of VOCs, the descent rate of
$O_3$ ($\Delta O_3$ (ppbv)/ $\Delta$VOCs (ppbv)) as a function of reduction percentage of VOCs were calculated (Figure
13 (b)). $O_3$ exhibited the highest dependence on BVOCs, with an average descent rate of $3.74 \pm 0.09$
ppbv/ppbv. Differing from the result in Figure 13 (a), diminishing alkenes could lead to decreasing of
$O_3$ by an average descent rate of $1.69 \pm 0.01$ ppbv/ppbv. On the contrary, reduction of $NO_x$ would lead
to increase of $O_3$, with an average rate of $1.29 \pm 0.21$ ppbv/ppbv (Figure S3). Although the descent rate
of $O_3$ turned to decrease and the sensitive of $O_3$ formation get into $NO_x$-limited regime when over 70%
of $NO_x$ were eliminated, it still causes net increase of $O_3$.

Although diminishing BVOCs seems to the most efficient way to restrain $O_3$ pollution, most of

BVOCs were emitted directly from plants and could not be easily controlled. Besides, huge number of
OVOCs (such as formaldehyde, aldehyde, methanol, ethanol, methyl vinyl ketone, methyl ethyl ketone,
etc.) could be directly emitted from anthropogenic processes or secondary formatted from the oxidation
of precursors (such as alkenes and aromatics), which complicates the control of OVOCs. Therefore,
considering the reduction potential and descent rate of $O_3$, more efforts are needed on the control of
alkenes and aromatics.

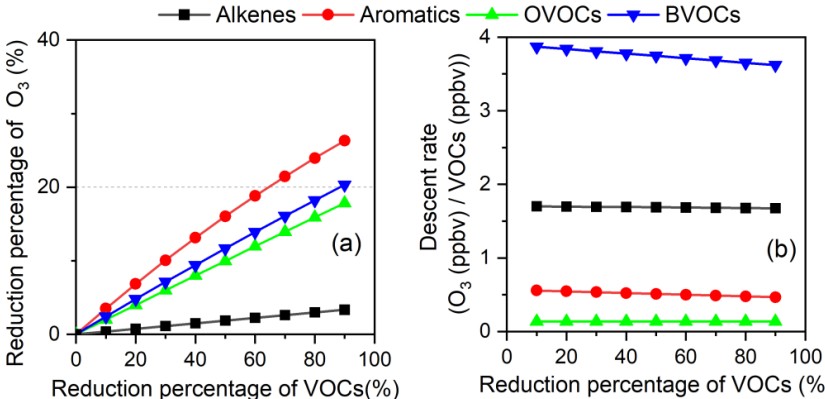


**Figure 13. Reduction percentage of $O_3$ as a function of reduction percentage of VOCs (a); descent rate of $O_3$ as a**
**function of reduction percentage of VOCs (b).**



## 4. Conclusions

After the outbreak of COVID-19, strict epidemic prevention measures have been adopted throughout China, leading to dramatic decrease in traffic volume and industrial activities. Affected by the decrease of number of vehicles on the road, non-essential industrial productivity, and associated pollutant emissions, most of the air pollutants (e.g., $PM_{2.5}$, $PM_{10}$, NO, $NO_2$, $SO_2$, and VOCs) dropped to a lower level during lockdown period (especially during Full-lockdown period). However, $O_3$ increased compared to that during the same period in 2019 in many urban areas of China. To figure out the reasons for this abnormal increase of $O_3$, the characteristics of $O_3$ precursors ($NO_x$, VOCs) during Pre-lockdown, Full-lockdown, and Partial-lockdown periods in Changzhou were analyzed. Although this study was conducted in single city of China, the representativeness of Changzhou guaranteed the applicability of the results the YRD region. Results suggested that the decrease of human activities during Full-lockdown period significantly suppressed the emissions of $NO_x$ and VOCs, which further lead to dramatic drop in the concentrations of most VOCs, especially aromatics. As a result, the $NO_x$/VOCs ratios dropped from 1.84 at Pre-lockdown period to 0.79 during Full-lockdown period. By deweathered calculation, we found that meteorology only contributed a minor positive (0.5 ppbv) value to the increase of $O_3$, whereas changes in precursor emissions led to 5.1 ppbv increase in $O_3$ concentrations during Full-lockdown period. To verify this result, a box model was used to simulate the formation of $O_3$. Results show that the AOC level during Full-lockdown was comparable to that during Pre-lockdown period, but the formation rate of $O_3$ was much higher during Full-lockdown period. By scenario analysis, we found the decrease of $NO_x$ and VOCs in Full-lockdown period dragged the formation of $O_3$ from VOC-sensitive regime to the junction of VOCs- and $NO_x$-limited regime, and the average simulated $MeanO_3$ in Full lockdown period could be 2.4 ppbv higher than that in Pre-lockdown period. Although the deweathered model and OBM model shows differences in the emission-derived change of $O_3$, the results together point out that the improper reduction of $NO_x$ and VOCs was the key reason for the abnormal increase of $O_3$ during Full-lockdown period in 2020. Overall, the outbreak of COVID-19 has caused devastation over the world. However, it provided an extreme experiment to



investigate the $O_3$ formation under strict emission control policies and provided insights into the policy
formulation for diminishing $O_3$ pollution in the YRD region. The data indicate that the concentrations
of VOCs and $NO_x$ have changed dramatically during the pandemic, a common situation also found in
other Chinese cities, and led to the switch of $O_3$ formation sensitivity. These results have a clear
indication that, in the future, more efforts should be paid on the reduction ratio of anthropogenic VOCs
and $NO_x$.

## Acknowledgement

This study was financially sponsored by the National Natural Science Foundation of China (grant
42075144, 41875161, 42005112), the Shanghai Science and Technology Innovation Plan (no.
19DZ1205007), the Shanghai Sail Program (no. 19YF1415600), the Shanghai International Science and
Technology Cooperation Fund (no. 19230742500).

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
