# Peer review of "Insights into the abnormal increase of ozone during COVID-19"

_Atmospheric Chemistry and Physics, 2021_

## Author Comment (AC1)

The manuscript entitled "Insights into the abnormal increase of ozone during COVID-19 in a typical urban city of China" by Kun Zhang et al. explored the drivers of elevated ozone concentration during COVID-19 lockdown. The manuscript provides valuable information for understanding ozone chemistry under rigorous emission reduction measures and efficiently directing ozone mitigation in the future. I would recommend publication if my following concerns are well addressed.

Response: We thank the reviewer for the positive and constructive comments and suggestions. All concerns have been carefully addressed. Below is our point-by-point response to each comment, marked in blue. Changes made to the main text are presented in green.

**General comments:**

(1) VOCs were measured by a PTR-TOF-MS in this study. However, this method cannot measure alkane and most alkene species, which will underestimate the ozone production and could mislead the diagnosis of ozone sensitivity regimes. Therefore, some uncertainty analysis regarding this deficiency is necessary.

Reply: Thanks for the constructive comment. Yes, the VOCs measured by PTR-TOF-MS do not include C2~C5 alkenes and alkanes. Unfortunately, we do not have traditional VOC observations during this period. To understand the possible underestimation of ozone production due to limitations of the measurement, we collected the observational data of C2~C5 alkenes and alkanes during the autumn of 2018 at the same site. We further performed simulations by including assumed diurnal variation of ethene, propene, butene, ethane, propane and butane which are key C2~C5 alkenes and alkanes at this site, in the model. During different runs, we added 0.5*[alkenes], 1*[alkenes], 1.5*[alkenes], or 2*[alkenes] in the model and calculate the uncertainties caused by alkenes. As shown in Figure 1 below, deficiency of C2~C5 alkenes and alkanes can lead to underestimation of $O_3$ during daytime. On average, adding 0.5~2 times alkenes or alkanes could lead to 1.65%~9.49% or 1.37~5.36% increase of

simulated $O_3$, respectively. Although the deficiency of C2~C5 alkenes and alkanes could bring in some uncertainty, the results of base case are still reliable for further analysis. Relevant description has been added in the revised manuscript, please refer to Page 22, Line 422-435:

Due to limitations in the observations, several issues should be noted in the application of the OBM model to evaluate the local chemistry in the present study. Firstly, deficiency of the observation of C2~C5 alkenes and alkanes could lead to underestimation of the simulated $O_3$. To understand the possible underestimation of ozone production due to limitations of the measurement, we collected the observational data of C2~C5 alkenes and alkanes during the autumn of 2018 at the same site. To analyze the uncertainties, we further performed simulations by including assumed diurnal variation of ethene, propene, butene, ethane, propane and butane which are key C2~C5 alkenes and alkanes at this site, in the model. On average, adding 0.5~2 times alkenes or alkanes could lead to 1.65%~9.49% or 1.37~5.36% increase of simulated $O_3$, respectively (Figure S7 and S8). In addition, the deficiency of C2~C5 has potential to cause uncertainty in $O_3$ formation potential. To quantify this impact, the EKMA analysis with the hypothetical diurnal variation of C2~C5 was also performed. The results suggested that the influence of the deficiency of C2~C5 alkenes and alkanes on the $O_3$ formation sensitivity is negligible (Figure S9). Therefore, although the deficiency of C2~C5 alkenes and alkanes could bring in some uncertainty, the results of base case are still reliable for further analysis.

[Figure]

**Figure 1 Sensitivity analysis of the influence of alkenes**

[Figure]

**Figure 2 Sensitivity analysis of the influence of alkanes**

(2) Temperature and solar radiation increase rapidly from Pre-lockdown period to Full-lockdown period, which could significantly contribute to the increase in ozone concentration during Full-lockdown period. This influence is not fully considered in the manuscript. Relevant analysis is also suggested to be included.

Response: Thanks for the comment. According to Figure 3 in the revised manuscript, the discrepancy between the $O_{3,Obs}$ during Pre-lockdown and Full-lockdown period could be partially attributed to changes of meteorological condition (11.74 ppb). Apart from the influence of meteorological condition, the $O_{3,Normal}$ in Full-lockdown period is still 0.64 ppb higher than that during Pre-lockdown period, which could only be attributed to the changes in emissions between the two periods. Relevant descriptions have been inserted to the revised manuscript, please refer to Page 10, Line 215-224:

It is obvious that the $O_{3,Obs}$ during Pre-lockdown period is much lower than that during Full-lockdown period in both years, which is partially attributed to the negative influence of meteorological conditions during Pre-lockdown period (Figure 3). This is consistent with the increasing temperature and solar radiation, which could significantly contribute to the increase of ozone concentration, from Pre-lockdown to Full-lockdown period. It should be noted that meteorology constrained $O_3$ concentrations by 3.9 ppbv during the Full-lock down period in 2019. Apart from the influence of meteorological condition, the $O_{3,Normal}$ in Full-lockdown

period in 2020 is still 1.46 ppbv and 0.64 ppb higher than that during Full-lockdown period in 2019 and that during Pre-lockdown period in 2020, indicating that improper decline of precursor emissions was possibly the key reason for the obvious increase of $O_3$ during Full-lockdown period in 2020.

**Specific comments:**

Line 28: "the observed $O_3$ "should be changed into "the increase in the observed $O_3$".

Response: We have revised this sentence as suggested.

Line 34-35: Here, the authors describe that the changes in precursor emissions (or $NO_x$/VOCs ratio) contributed 2.4 ppbv to the $O_3$ increase, which is inconsistent with 5.1 ppb in lines 27-28. Please double check it.

Response: We are sorry for the improper description. We used deweathered method and box model to estimate the influence of emission changes, respectively. The result from deweathered method was described in Line 27-29, while the result from box model was shown in Line 34-35. To avoid misdirection, we have revised Line 34-35 into:

Additionally, box model results suggested that the decrease in NOx/VOCs ratio during Full-lockdown period was supposed to increase the MeanO$_3$ by 2.4 ppbv.

Line 58-59: also include actinic flux in meteorological conditions and cite the papers (1, 2).

Response: We have included actinic flux in meteorological conditions and cited the papers as suggested.

References:

Wang et al., The impact of aerosols on photolysis frequencies and ozone production in Beijing during the 4-year period 2012–2015. Atmos. Chem. Phys. 19, 9413-9429 (2019).

Wang et al., Exploring the drivers of the increased ozone production in Beijing in summertime during 2005–2016. Atmos. Chem. and Phys. 20, 15617-15633 (2020).

Line 175-176: The influence of RH on ozone is very complicated. Higher humidity is conducive to OH production and thus likely increase $O_3$ production. I suggest to add some references about RH influence here and simulate the influence of RH on ozone by box model.

Response: We agree that the influence of RH on $O_3$ is complicated, which are non-linearly related. We have added some references about RH influence. In addition, we have added sensitivity analysis to quantify the influence of RH by increasing or decreasing RH by 10%, and the results are exhibited in Figure 3. On average, reducing RH by 10% leads to 0.28% increase of the simulated $O_3$, and this influence dropped to -0.35% when RH was 5% lower than the base case. When increasing RH by 5% or 10%, positive influence on the simulated $O_3$ was found. Therefore, we have revised the descriptions in the manuscript, please refer to Page 9, Line 198-201:

The relatively higher T was in favor of $O_3$ formation during the Full-lockdown period in 2020. As for RH, the influence on $O_3$ is nonlinear (Zhang et al., 2020), and based on our sensitivity test, lower RH could lead to decrease or increase of O3 concentration (Figure S2).

[Figure]

**Figure 3. Sensitivity analysis of the influence of RH**

Line 177: please explain why RH>70% can be an indicator of adverse weather conditions.

Response: We are sorry for this misleading information, RH>70% is not an indicator of adverse weather conditions. Therefore, relative description has been removed.

Line 192: What does the $r^2$ represent? You should state out it in Section 2.4.

Response: The $R^2$ represents the determining coefficient of the model and a $R^2$ close to 1 means the model can reproduce the observation well. We have stated out $R^2$ in section 2.4. Please refer to Line 160-162:

The results suggest that the highest coefficient of determination ($R^2$, 0.84) was obtained when ntree, nsample and minimal node size was set as 300, 300, and 5, respectively (Table S1 and S2).

Line 195: I think "$O_{3,Obs}$" should be "$O_{3,Normal}$" here.

Response: Thanks for pointing this out. Yes this should be "$O_{3,Normal}$". We have revised it in the latest version of manuscript.

Figure 3 and Figure 12: In figure 3, $O_{3,Normal}$ during Full-lockdown period is higher than that during Pre-lockdown period by 12 ppb. However, the corresponding value is only 2.4 ppb. Please explain this inconsistency.

Response: We apologize for the mistake in the original figure 3, which used the predicted $O_3$ instead of the weather-normalized $O_3$ data. We have updated figure 3 with the correct result, and the figure shows the $O_{3,Normal}$ during Full-lockdown is 0.7 ppbv higher than that during Pre-lockdown period, which is close to the value in figure 12 (2.4 ppb).

Figure 3: My understanding is that the deweathered method normalizes the influence of meteorological factors on the difference between the same periods in different years. Were meteorological factors between different periods also normalized? The authors should clearly explain this in Section 2.4. This is important to figure out the influence of meteorological factors on ozone increase during Full-lockdown period compared to pre-lockdown period.

Response: The meteorological factors between different periods were not normalized. They were sampled to predict concentrations many times (and aggregated) to calculate the normalized time series. The corresponding description has been revised in Section 2.4. Please refer to Line 164-169:

Training of the models was conducted on 80% of the input data and the other 20% was withheld from training. To avoid the disadvantage of overfitting during the training of RF, a process called bagging (or bootstrap aggregation) was adopted. Bagging results in new, sampled set called out-of-bag (OOB) data. A decision tree is then grown on the OOB data. Therefore, all the decision trees are grown on different observations and avoid the overfitting (Grange and David (2019)).

Line 214-215: I suggest to at least give some evidences that the decrease in VOC is due to the decrease in industrial activities and traffic volume. Besides industrial activities and traffic volume, solvent usage is also an important source of VOC.

Response: Thanks for the helpful suggestion. We summarized the electricity consumption of key industries in Changzhou during our observation, and calculated the corresponding VOC emissions using the following equation. We also collected the traffic volume data during the observational period.

$$E_o = \sum_{i=1}^{n} \frac{E_{pi}}{S_{pi}} \times S_{oi}$$

where $E_o$ (unit: t) is the total daily VOC emission from industrial sources during the observation; $E_{pi}$ (unit: t) and $S_{pi}$ is the daily VOC emissions and electricity consumption of the $i^{th}$ industry during the second national pollution census, respectively; $S_{oi}$ is the daily electricity consumption during our observation; n is the number of industries. The VOC emissions data and the electricity consumption data was obtained from the second national pollution census and Atmospheric Information Platform of Changzhou (http://58.216.50.59/), respectively. It is

clearly shown that the time series of industrial VOC emissions and traffic volume showed similar trend during the observation, suggesting that the lock-down policy strongly influence industry and traffic simultaneously. In addition, area source like solvent usage is also an important source of VOCs, which is prohibited during the Lock-down period. To prove that the decrease of VOCs during Full-lockdown period is caused by changes in human activities (industries, traffic, and solvent use, etc) the variation of typical industry-derived VOC (styrene) and traffic/industry-derived VOC (benzene toluene and xylene) are presented in Figure 4. In addition, the relevant description has been added in the revised manuscript, please refer to Page 11, Line 235-236:

This is proved by the trend of traffic volume, VOCs emission and traffic/industrial-derived VOCs (Text S1 and Figure S3).

[Figure]

**Figure 4 Time series of industrial-derived VOCs emissions, traffic volume, and key VOC tracers.**

Line 266-268: The expression is ambiguous here. Acetaldehyde and formaldehyde don't belong to aromatics.

Response: We are sorry for this mistake. Text has been revised:

Among VOCs, xylene exhibited the maximum OFP value ($68.6 \pm 59.3$ μg/m$^3$), followed by acetaldehyde ($28.8 \pm 6.4$ μg/m$^3$), toluene ($25.7 \pm 20.1$ μg/m$^3$) trimethylbenzene ($25.4 \pm 15.8$ μg/m$^3$), and formaldehyde ($22.7 \pm 9.1$ μg/m$^3$).

Line 273-274: "As for alkene, this could be explained by their chemical reactivities, which led to the fast degradation after emission." I don't agree with this statement as aromatics tend to have similar chemical reactivity as alkenes.

Response: We agree that the reactivities for some alkenes and aromatics are similar, but it may not be suitable for our observation. During this observation based on PTR, the most abundant alkenes are 1-hexene and isoprene, with the $k_{OH}$ of 37 and $100 \times 10^{-12}$ $cm^3$ molecule$^{-1}$s$^{-1}$, respectively. As for aromatics, the most abundant species are benzene, toluene and xylene, with the $k_{OH}$ of 1.22, 5.63 and 17 $cm^3$ molecule$^{-1}$s$^{-1}$, respectively. Therefore, the reactivities of the observed alkenes are much higher than that of aromatics during this period in the study area. Hence, the relatively smaller change of OFP form alkenes could be explained by their chemical reactivities. To avoid misunderstanding, we have revised descriptions into:

During the observation, the most abundant alkenes measured by PTR-TOF-MS are 1-hexene and isoprene, with the $k_{OH}$ of 37 and $100 \times 10^{-12}$ $cm^3$ molecule$^{-1}$ s$^{-1}$, respectively,which are much higher than that of the most abundant aromatics (1.22, 5.63, and 17 $cm^3$ molecule$^{-1}$ s$^{-1}$ for benzene, toluene, and xylene, respectively). The fast degradation of these alkenes could attribute to the small relatively smaller change of OFP from alkenes.

Line 277-278: This could also be due to enhanced solar radiation and temperature from January to March.

Response: We totally agree. The corresponding description has been revised, please refer to Line 303-304:

which could be attributed to the higher AOC, enhanced solar radiation and temperature during Partial-lockdown period.

Line 281-282: "This result suggests that the VOCs in Partial-lockdown should produce less O3 than that in Pre-lockdown, and Partial-lockdown period". This is misleading. First, the former

"Partial-lockdown" should be Full-lockdown. Second, the MIR that you calculate here refers to the ability of VOC species composition to produce ozone, is it correct? Thus, I suggest to further explain the concept of MIR and change this sentence into "This result suggests that VOC species composition in Full-lockdown is more conducive to ozone formation………". In addition, please specify each dot represent 1-hour average or 24-hour average in Figure note.

Response: Thanks for pointing this out. The former "Partial-lockdown" has been revised to "Full-lockdown". Yes, the MIR calculated here refers to the ability of VOC species composition to produce ozone. To avoid misleading, we have explained the concept of MIR, and this sentence has been revised as suggested, please refer to Lin 304-311. In addition, we have specified that the dot are 1-hour averaged in the note of Figure 7.

To compare the average reactivity of VOCs during different periods, we calculated the mean MIR, derived by dividing the total OFP by total VOC concentration, in each period. A higher MIR means stronger capability of VOCs to produce ozone. As shown in Figure 7, the average MIR during Pre-lockdown, Full-lockdown, and Partial-lockdown period was 3.85, 3.53 and 3.68 (g $O_3$/g VOC), respectively. This result suggests that VOC species composition in Full-lockdown is more conductive to ozone formation than that in Pre-lockdown, and Partial-lockdown period. However, the formation of $O_3$ was sensitive to the ratio of $NO_x$/VOCs and meteorological conditions, which can be significantly different in each period.

Line 304: "Feb 14[th]" should be "Feb 1[st]".

Response: We have revised this sentence as suggested.

Line 325-327: I suggest to explain the reason why OVOC kept stable among the three cases, which is inconsistent with the remarkable difference in measured OVOC among the three periods as you shown in Figure 4.

Response: Thanks for the suggestion. The original description is inaccurate. After double check

of the data, we found that during the three periods, the $k_{OH}$ and concentration of OVOC exhibited similar "U-shaped" trend, with the minimum during Full-lockdown period. To avoid misleading, this sentence has been revised:

As $k_{OH}$ from OVOC, it shared same trend as OVOC concentration, which reached the minimum value (5.56 s$^{-1}$) during the Full-lockdown period.

Line 330-355: PTR-TOF-MS is unable to measure alkanes and most alkenes, which could influence the diagnosis of ozone sensitivity to precursors. Lower VOCs concentrations lead to more VOC-limited regime. I suggest to provide uncertainty analysis about it.

Response: Thanks for the comment. To investigate the influence of the deficiency of C2~C5 alkenes and alkenes, we used the hypothetical diurnal variation of ethene, propene, butene, ethane, propane and butane as mentioned above and conducted EKMA analysis. Generally, adding C2~C5 alkenes and alkanes in the model would lead to slight increase of the simulated $O_3$, and could not obviously change the shape of $O_3$ isopleth (Figure 5). Therefore, the influence of the deficiency of C2~C5 alkenes and alkanes on the $O_3$ formation sensitivity is negligible. It should be noted that, this sensitivity analysis is based on the "hypothetical" diurnal variation of C2~C5 alkenes and alkanes, which would bring in uncertainty. We hope a wider range of VOCs would be monitored simultaneously in future field campaign and avoid this dificiency. The relative description has been added in the revised manuscript, please refer to Page 22, Line 431-437:

In addition, the deficiency of C2~C5 has potential to cause uncertainty in $O_3$ formation potential. To quantify this impact, the EKMA analysis with the hypothetical diurnal variation of C2~C5 was also performed. Generally, adding C2~C5 alkenes and alkanes in the model would lead to slight increase of the simulated $O_3$, and could not obviously change the shape of $O_3$ isopleth (Figure S9). Therefore, the influence of the deficiency of C2~C5 alkenes and

alkanes on the O$_3$ formation sensitivity is negligible. It should be noted that, this sensitivity analysis is based on the "hypothetical" diurnal variation of C2~C5 alkenes and alkanes, which would bring in uncertainty. We hope a wider range of VOCs would be monitored simultaneously in future field campaign and avoid this deficiency.

[Figure]

**Figure 5. MeanO$_3$ isopleth with (left) and without (right) hypothetical diurnal variation of C2~C5 alkenes and alkanes. The colored circles, triangles, and rectangles represent the daily average**

Figure 2: The legend of different parameters at the top of the Figure should be placed in corresponding sub-panels. Besides, the legend of ozone and TVOC is not given at present.

Response: The legend of each parameter was placed in corresponding sub-panels. In addition, the legend of O$_3$ and TVOC has been revised to be more conspicuous.

Section 2.2: How were photolysis frequencies been considered in the model? Were they constrained by photolysis measurements or calculated by a radiative transfer model (e.g., TUV)? If they were calculated, what about the uncertainty compared to the real condition? And what would be the influence on the afterwards data analysis?

Response: The photolysis frequencies (J values) were calculated as a function of solar zenith angle, altitude using lookup tables, calculated using the Tropospheric Ultraviolet and Visible (TUV) model, and relative description has been added in the revised manuscript. Since no observational data of J value is available, it is unable to calculate the uncertainty. Here, we

only analyze the sensitivity of the simulated $O_3$ to J values by increasing or decreasing the photolysis rates by 10% and 20%. Results show that the simulated $O_3$ could decrease or increase by 25.14% or 21.73%, respectively, when photolysis rates were decreased or increased by 20% (Figure 6). In addition, the J values, which directly or indirectly influence the recycling of $RO_x$, could lead to uncertainty to the calculation of AOC and $k_{OH}$. The relative changes in AOC and $k_{OH}$ by 1% changes in J values was 1.07%/% and 0.14%/%, respectively. Therefore, synchronously measurement of J values is recommended for future field campaign. Relative description has been added in the revised manuscript. Please refer to Page 23, Line 435-444:

Secondly, the photolysis frequencies (J values) were calculated as a function of solar zenith angle, altitude using lookup tables, calculated using the Tropospheric Ultraviolet and Visible (TUV) model, which could lead to uncertainty in the simulation of $O_3$. Hence, we analysis the influence of J values by increasing or decreasing the photolysis rates by 10% and 20%. Results showed that the simulated $O_3$ could decrease or increase by 25.14% or 21.73%, respectively, when photolysis rates were decreased or increased by 20% (Figure S10). In addition, the J values, which directly or indirectly influence the recycling of $RO_x$, could lead to uncertainty in the calculation of AOC and $k_{OH}$. Based on above sensitivity analysis, we found the relative changes in AOC and $k_{OH}$ by 1% changes in J values was 1.07% and 0.14%, respectively. Therefore, the J values is recommended to be measured during future observations.

[Figure]

**Figure 6. Uncertainty analysis of J-value**

Line 375-376: "underestimation of ozone sensitivity to alkanes and alkenes" should be "underestimation of ozone production from alkanes and alkenes".

Response: We have revised this sentence as suggested.

---

## Author Comment (AC2)

This study targets an important question, what causes the ozone increase during lockdown despite substantial decrease in anthropogenic emissions? By applying some statistical approaches, the authors decouple the effects of changing meteorology and emission on ozone formation, and reported that changes in emissions causes a 5 ppb increase in ozone during the lockdown, where changes in meteorology conditions only increase ozone by 0.5 ppb. Further, it is shown that the ozone formation shifts from a VOC-limited regime before lockdown to the conjunction of NOx- and VOC-limited regime, which increase ozone formation. Overall, the scope of this study fits the journal. I recommend publication after major revisions.

Response: We thank the reviewer for the positive and constructive comments. Below is our point-by-point response to each comment, marked in blue. Changes made to the main text are presented in green.

Major Comments

Several statistical methods are applied in the study, but it is not clearly stated why they are selected? For example, why Sen's slope is used rather than a simple linear regression? There is a myriad of machine learning algorithms, so that the rationale behind each selection should be discussed. For example, Sen's slope is a robust slope and less susceptible to outliers. Further, current description of deweathered model lacks details. What does the model do? If I understand correctly, it takes several parameters as inputs and use random forest to predict O3 concentration, right?

Response: Thanks for the comment. The Sen's slope is selected since it is insensitive to outliers, and does not require a normal distribution of residuals. The deweathered model is used to remove the influence of meteorological conditions and obtain the hypothetical $O_3$ concentrations under the same "normalized weather condition", so that we can discuss whether the abnormal increase of $O_3$ is due to meteorological condition or changes in emissions. More details of the Sen's slope and deweathered model has been inserted into the

 The Sen's slope is selected since it is insensitive to outliers, and does not require a normal distribution of residuals.

 The observed concentrations of $O_3$ could be influenced by meteorological conditions, emissions and/or chemistry. The emissions and chemistry are being treated together and separated from meteorology by the deweathered approach based on the random forest (RF). Hourly data of Unix date (number of seconds since 1970-01-01), Julian day, weekday, hour of day, wind speed (WS), wind direction (WD), temperature (T), relative humidity (RH), and pressure (P), which are available during the whole observation, were used for the deweathered calculation of $O_3$. The missing data was replaced by linear interpolation. Training of the models was conducted on 80% of the input data and the other 20% was withheld from training. To avoid the disadvantage of overfitting during the training of RF, a process called bagging (or bootstrap aggregation) was adopted. Bagging results in new, sampled set called out-of-bag (OOB) data. A decision tree is then grown on the OOB data. Therefore, all the decision trees are grown on different observations and avoid the overfitting (Grange and David, 2019). To determine the value of number of trees (ntree), number of samples (nsample), and the minimal node size, a series of random forests were performed under different choices of ntree, nsample, and minimal node size. Results suggest that the highest coefficient of determination ($R^2$, 0.84) was obtained when ntree, nsample and minimal node size was set as 300, 300, and 5, respectively (Table S1 and S2). More details of this model could be found in the study of Grange and David (2019). The uncertainty of the deweathered model is obtained by growing 50 random forest models with the hyperparameters described above, which is the same method as Grange and Carslaw (2019). The mean and standard error of the predicted $O_3$ concentrations is presented in Figure S1, and results of the model are stable during the 50 runs.

The interpretation of $O_{3,met}$ and $O_{3,emission}$ is confusing, partly because of lack of details in describing the stats methods. To the reader, the difference between observed $O_3$ and weather-normalized $O_3$ represents the influence of changing emission, as weather-normalized $O_3$ takes into account the variation in $O_3$. The difference in observed $O_3$ between different years does not represent the influence of emission, because the meteorology between different years is different. Such interpretation will fundamentally change the conclusion on this manuscript as well as the conclusions from the box model. Please clarify.

Response: Thanks for the comment. The difference between observed $O_3$ ($O_{3, Obs}$) and weather-normalized $O_3$ ($O_{3, Normal}$) represent the influence of meteorology, which is consistent with the definition in Li et al. (2021). The differences in $O_{3,Normal}$ among different years represent the influence of emissions, since the $O_{3,Normal}$ has already removed the influence of meteorological conditions. To avoid misunderstanding, relevant descriptions has been added in the revise manuscript. Please refer to Page 8, Line 177-181:

The differences in observed $O_3$ concentrations ($O_{3,Obs}$) and deweathered $O_3$ concentrations ($O_{3,Normal}$) were regarded as the concentrations contributed by meteorology ($O_{3,Met}$), which is consistent with the definition in Li et al. (2021). Correspondingly, the differences in $O_{3,Normal}$ concentrations in different periods represent the influence of emissions, since the $O_{3,Normal}$ has already removed the influence of meteorological conditions.

The discussions on ozone formation potential (OFP) can be reconstructed in a more meaningful way. Mainly, it should be clearly stated that OFP does not indicate $O_3$ concentration. With this premise, there is no need to discuss "consistency" or "inconsistency" between the two (Line 282). In other words, OFP is not helpful to answer the O3 question in the manuscript.

Response: We agree that OFP only gives the $O_3$ formation potential by the observed VOCs and it does not directly indicate $O_3$ concentration changes. But the discussion of OFP can be

a reference for the comparison of the reactivity of VOCs in each period. Therefore, we have revised the relevant description, please refer to Page 16, Line 304-310:

To compare the average reactivity of VOCs during different periods, we calculated the mean MIR, derived by dividing the total OFP by total VOC concentration, in each period. A higher MIR means stronger capability of VOCs to produce ozone. As shown in **Error! Reference source not found.**, the average MIR during Pre-lockdown, Full-lockdown, and Partial-lockdown period was 3.85, 3.53 and 3.68 (g $O_3$/g VOC), respectively. This result suggests that VOC species composition in Full-lockdown is more conductive to ozone formation than that in Pre-lockdown, and Partial-lockdown period.

The reliability of the box model results is compromised by the fact that the modeled $O_3$ during full lockdown (29 ppb) is lower than that during partial lockdown (32ppb), which contrasts the observation.

Response: We think there could be misunderstandings. According to the results of the box model and Figure 8 in the revised manuscript, the modeled daytime $O_3$ concentration during Full-lockdown (36.4 ppbv) is higher than that during Partial-lockdown (33.3 ppbv), which is consistent with the observation.

Minor Comments

Line 167 is a confusing sentence.

Response: The original sentence is confusing; hence we have revised:

It should be noted that the decreasing ratio of VOC/$NO_x$ is around 1.75, suggesting that the lockdown policy has stronger influence on $NO_x$ emissions than VOC emissions.

Line 257. "Vary", not "varies".

Response: Revised.

Line 281. It is "full-lockdown", not "partial-lockdown".

Response: Revised.

Line 299. What does "AOC" represent?

Response: AOC represents the atmospheric oxidation capability, and the relative description has already been declared in line 129: According to the definition of atmospheric oxidation capability (AOC)....

---

## Author Comment (AC3)

The subject manuscript by Zhang et al. presents an analysis of ambient measurements from the Yangtze River Delta assessing the observations of increased ozone ($O_3$) levels during the COVID-19 lockdown period. Results are presented for three periods: pre lockdown, full lockdown, and partial lockdown and are compared with the same time periods in 2019. The authors seek to understand the relative importance of precursor volatile organic compounds (largely grouped by compound class), nitrogen oxides (NOx), ambient reactivity/oxidation capacity, and meteorology in determining ozone levels. The authors motivate the study well, and adequately convey the importance of measurements and analysis in this region. However, the methodology is not sufficiently described to support the results and conclusions, and the results and conclusions are somewhat difficult to follow as written. Specific questions, comments and suggestions on these points follow below. It is recommended that this manuscript be reconsidered for publication after major revisions.

Response: We thank the reviewer for the helpful comments and suggestions. We have carefully addressed all the comments and suggestions. Below is our point-by-point response to each comment, marked in blue. Changes made to the main text are presented in green.

Technical Comments:

line 34, lines 173-176: Using "supposed to" in this sentence does not necessarily reflect the complexity of $O_3$ formation. The prior sentence suggests that during the full lockdown period, the region shifted to a NOx-limited regime. Thus, it may be expected that a greater decrease in NOx relative to VOCs would lead to a decrease in $O_3$. However, there is also the role of NOx titration, in which decreasing NOx can lead to an increase in $O_3$. Later in the manuscript, this phrasing is repeated and it suggests that the authors are not necessarily referring to the NOx regime in the abstract, but the influence of meteorology (specifically T, RH). Again, this is not clear in the abstract, and oversimplified as written. What is the mechanism by which RH affects $O_3$ formation? How sensitive is $O_3$ to RH?

Response: (1) We agree that using "supposed to" in this sentence does not necessarily reflect the complexity of $O_3$ formation. The obvious increase of $O_3$ during the Full-lockdown period was caused by the joint effect of meteorology, changes in emissions and chemistry. Therefore, we have revised "supposed to" to "in favor of". From Pre-lockdown to Full-lockdown period, the average $NO_x$ and VOCs concentrations decreased by 62.6% (20.0 ppbv) and 32.2% (10.5 ppbv), respectively, while $O_3$ concentration increased by 67% (12.4 ppbv). This is attributed to the fact that the $O_3$ formation was VOC-limited during Pre-lockdown period, and more abatement of $NO_x$ than VOCs would lead to increase of $O_3$. If the over 80% of $NO_x$ and over 45% of VOCs were eliminated, like the case of $16^{th}$ Feb and $17^{th}$ Feb, the $O_3$ formation would switch to the $NO_x$-limited regime and lead to decrease of $O_3$.

Therefore, we have revised relevant description in the abstract and discussion, please refer to Page 2, Line 32-34:The $NO_x$/VOCs ratio dropped dramatically from 1.84 during Pre-lockdown to 0.79 in Full-lockdown period, which switched $O_3$ formation from VOCs-limited regime to the boundary of $NO_x$- and VOC-limited regime.".

(2) The mechanism by which RH affects $O_3$ formation is very complicated. Higher humidity is conducive to OH production and thus likely increase $O_3$ production. We have added some references about RH influence here and simulated the influence of RH on ozone by box model. Sensitivity analysis has been performed to reveal this effect by increasing or decreasing RH by 10%, and the results are exhibited in Figure 1. On average, decreasing RH by10% leads to 0.28% increase of the simulated $O_3$, and this influence dropped to -0.35% when RH was 5% lower than the base case. When increasing RH by 5% or 10%, positive influence on the simulated $O_3$ was found.

[Figure]

**Figure 1. Sensitivity analysis of the influence of RH on simulated O₃**

We have revised the manuscript accordingly, please refer to Page 9, Line 198-201:

The relatively higher T was in favor of O₃ formation during the Full-lockdown period in 2020. As for RH, the influence on O₃ is nonlinear (Zhang et al., 2020), and based on our sensitivity test, lower RH could lead to decrease or increase of O₃ concentration (Figure S2).

Similar to the use of "supposed to", the authors need to clarify "improper" decline (line 202) and "abnormal" increase of O3 (line 203).

Response: Thanks for the helpful suggestion. The "abnormal" represent the obvious higher O₃ concentration in the Full-lockdown period in 2020 than that during the same time in 2019. To avoid misunderstanding, we have replaced "abnormal" with "obvious", and revised relative description in the manuscript, please refer to Page 9, Line 192-195:

It should be noted that, compared to Full-lockdown period in 2019, the mean O₃ concentration in 2020 is obviously higher (5.5 ppbv, **Error! Reference source not found.**). Meanwhile, the

average $O_3$ concentrations in Full-lockdown period in 2020 was 67% higher than that during Pre-lockdown period in 2020.

PTR-TOF-MS measurements (p. 5): Does the Jensen et al. companion paper address losses in the inlet and to the filter? How might these losses affect the results of the analysis presented here? The authors do not need to provide all of the details presented in Jensen et al., but should summarize the main findings, including any limitations, that are relevant to the analysis presented in this manuscript.

Response: The Jensen et al. companion paper did not address the loss in the inlet and the filter. During the observation, a 3-m long PTFE tube was used as inlet, and no strong loss was supposed. In addition, standard gases were used routinely for the calibration. Therefore, we believe the results from PTR-TOF-MS is reliable. Additionally, the main findings and limitation of Jensen et al. (2021) that are relevant to this manuscript have been added in the revised manuscript, please refer to Page 12, Line 236-238:

In addition, Jensen et al. (2021) found the VOC emissions from most industries in Changzhou share the same "U-shape" trend as our study."

Trend analysis: The authors state that the MK non-parametric test is recommended by the WMO. The authors should provide some additional detail here. What does the WMO recommend this test for? Under what conditions? What are the limitations/requirements for applicability in the context of this work? How is serial correlation applicable to the PTR-TOF-MS measurements of individual VOCs? What details of Pathakoti et al. and Alhathloul et al. are relevant here?

Response: According to Adeloye and Montaseri (2002) and WMO (1998), the Spearman Rank Order Correlation (SROC) instead of MK test is recommended to investigate the long-term trend of flow volume, and we have revised this sentence in the manuscript. We select MK test

because it is a non-parametric statistical method and doesn't require any assumptions regarding the probability distribution of the data. The MK test has a limitation that the input data should have no serial correlation. The serial correlation of individual VOC has been tested by the "feasts" R package, and no serial correlation is found for each individual VOC. The paper of Pathakoti et al. and Alhathloul et al. gives the details of the calculation MK trend test. Relevant description has been revised, please refer to Page 7, Line 145-148:

By using the "feast" R package, no obvious serial correlation of individual VOC is found. Therefore, the observed VOC data is suitable for MK test. Detailed description and the calculation formula of MK trend test could be found in the study of Pathakoti et al. (2021) and Alhathloul et al. (2021).

Deweathered model: While details of the VOC measurements are somewhat lacking, and more so for the trend analysis, this section is entirely lacking of sufficient detail (and is not, as the authors note on line 191, described in section 2.4). The authors should consider that the information in the manuscript needs to be sufficient such that the results can be reproduced. Further, it is difficult to assess the robustness of the results when sufficient details about the methodology are not provided. What are the uncertainties of the approach? Are data available for all parameters over all time periods? How are missing data handled? How were the model parameters determined (number of trees, minimal node size, and number of samples)? How sensitive are the results to these model parameters?

Response: We agree that the details of deweather analysis is insufficient. All parameters used in this model is available over all period and a small amount of missing data is replaced by linear interpolation. The number of trees (ntree), number of samples (nsample) and minimal node size were selected by the sensitive analysis (Table 1 and 2). According to Table 1 and 2, when ntree, nsample and minimal node size were chosen as 300, 300, and 5, respectively, the

R$^2$ of the deweather model is the highest. In addition, the R$^2$ of the deweather model is not sensitive to the choose of ntree, nsample, and minimal node size (Table 1 and 2). The uncertainty of the deweather model is obtained by growing 50 random forest models with the nree, nsample and minimal node size chosen as 300, 300, and 5, respectively, which is the same method as Grange and Carslaw (2019). The mean and standard error of the predicted O$_3$ concentrations is shown in Figure 2, and results of the model are stable during the 50 runs.

We have added the details of the deweather model in the revised manuscript, please refer to Page 8, Line 160-177:

Hourly data of Unix date (number of seconds since 1970-01-01), Julian day, weekday, hour of day, wind speed (WS), wind direction (WD), temperature (T), relative humidity (RH), and pressure (P), which are available during the whole observation, were used for the deweathered calculation of O$_3$. The missing data was replaced by linear interpolation. Training of the models was conducted on 80% of the input data and the other 20% was withheld from training. To avoid the disadvantage of overfitting during the training of RF, a process called bagging (or bootstrap aggregation) was adopted. Bagging results in new, sampled set called out-of-bag (OOB) data. A decision tree is then grown on the OOB data. Therefore, all the decision trees are grown on different observations and avoid the overfitting (Grange and David, 2019). To determine the value of number of trees (ntree), number of samples (nsample), and the minimal node size, a series of random forests were performed under difference choice of ntree, nsample, and minimal node size. The results suggest that the highest coefficient of determination (R$^2$, 0.84) was obtained when ntree, nsample and minimal node size was set as 300, 300, and 5, respectively (Table S1 and S2). More details of this model could be found in the study of Grange and David (2019). The uncertainty of the deweather model is obtained by growing 50 random forest models with the hyperparameters described above, which is the same method as

Grange and Carslaw (2019). The mean and standard error of the predicted $O_3$ concentrations is shown in Figure S1, and results of the model are stable during the 50 runs."

[Figure]

**Figure 2. The mean and standard error of predicted $O_3$ concentrations.**

**Table 1 $R^2$ of the deweather model with different choose of ntree and nsample.**

| ntree\nsample | 100 | 200 | 300 | 400 | 500 |
|---|---|---|---|---|---|
| 100 | 0.852 | 0.853 | 0.853 | 0.852 | 0.852 |
| 200 | 0.855 | 0.855 | 0.856 | 0.855 | 0.855 |
| 300 | 0.856 | 0.857 | 0.858 | 0.856 | 0.856 |
| 400 | 0.857 | 0.857 | 0.857 | 0.857 | 0.856 |
| 500 | 0.857 | 0.857 | 0.857 | 0.857 | 0.857 |

**Table 1 Influence of the choose of minimum node size on $R^2$ of the deweather model.**

| minimal node size | 1 | 2 | 3 | 4 | 5 |
|---|---|---|---|---|---|
| $R^2$ | 0.860 | 0.857 | 0.858 | 0.858 | 0.859 |
| minimal node size | 6 | 7 | 8 | 9 | 10 |
| $R^2$ | 0.855 | 0.853 | 0.852 | 0.851 | 0.849 |

In general, in the results and discussion, it is often difficult to follow whether the authors are describing results between the three periods in 2020, or between given periods in 2019 and 2020. It is recommended that the authors try to more clearly differentiate these comparisons.

Response: Thanks for the helpful suggestion. The comparisons in the results and discussion have been clearly differentiated as suggested.

line 214: TVOC dropped to 22.19 ppb from what mixing ratio?

Response: This sentence has been revised to:

Full-lockdown period, the TVOC dropped to $22.19 \pm 7.9$ ppbv from $32.78 \pm 13.81$ ppbv, which was mainly affected by the decrease in industrial activities and traffic volume.

lines 218 and 234: The authors use "interesting" in these sentences, but it is not clear what is interesting about the observations as presented. The higher mixing ratios due to lower boundary layer heights (line 218) is a common observation, and the lower values of transportation-associated VOCs (line 234) during the lockdown is expected (and has been reported previously).

Response: Thanks for the suggestion, we have removed "interesting" in these sentences.

In line 233, is the decreasing trend based on the Z-score or Q value? Were these metrics consistent? Why or why not? It might be useful to include the Z-scores and Q values for all compounds in the SI.

Response: The decreasing trend is based on Q value or Z score, which has been mentioned in the manuscript. The Z value and Q value are consistent because both indicate the trend of a time series. But the Q value is usually used for the quantification of the rate of the trend. The Z score and Q values for all compounds are included in the SI.

line 224-226: Can the authors be more quantitative about how many of the measured VOC species shown exhibited this U-shape pattern and then explicitly list those VOCs that didn't? It is a little contradictory to say "most" and then "except for several".

Response: Thanks for the helpful suggestion. We have revised relative description, please refer to Page 13, Line 248-259:

Total 42 VOC species exhibited an 'U' shape trend during the whole observation, while formaldehyde (HCHO) and methanol showed an obvious increasing pattern.

line 269: Is it expected that biogenic emissions would be the dominant source of $O_3$ in this region?

Response: This site is in the urban area of Changzhou city, where the VOCs are dominated by anthropogenic sources. Hence, the dominant source of $O_3$ in this region is not expected to be biogenic emissions. To avoid misleading, we have removed this sentence.

lines 273-275: This discussion of alkenes is not clear as written. The chemical reactivities of compounds is tied to their oxidation formation potential (and is not independent of).

Response: Thanks for the comment. The discussion of alkenes has been revised, please refer to Page 17-18, Line 296-299:

However, the OFP of alkenes and OVOCs only decreased by 8.9 and 22.5 $\mu g/m^3$, respectively. During the observation, the most abundant alkenes measured by PTR-TOF-MS are 1-hexene and isoprene, with the $k_{OH}$ of 37 and 100 $\times 10^{-12}$ $cm^3$ molecule$^{-1}$ s$^{-1}$, respectively, which are much higher than that of the most abundant aromatics (1.22, 5.63, and 17 $cm^3$ molecule$^{-1}$ s$^{-1}$ for benzene, toluene, and xylene, respectively).

line 279: How are the MIR values calculated? What are they dependent on? Is it expected that the MIR would be reflective of the different NOx/VOC regimes? I'm not sure this is the case.

Response: The mean MIR was calculated by dividing the total OFP by the total concentration of VOC, this parameter depends on the individual MIR and concentration of each VOCs. The mean MIR could not reflect the different $NO_x$/VOC regimes, while it can represent the ability of VOC species composition to produce ozone. The relevant description has been added in the revised manuscript, please refer to Page 17, Line 304-307:

To compare the average reactivity of VOCs during different periods, we calculated the mean MIR, derived by dividing the total OFP by total VOC concentration, in each period, and a higher MIR means stronger capability of VOCs to produce ozone.

line 286: Was $NO_x$ eliminated (which suggests some chemical/physical removal)? Or were the emissions reduced to a greater extent that VOC emissions?

Response: To avoid misleading, the sentence has been revised to "suggesting more $NO_x$ was reduced than VOCs during Full-lockdown period".

line 326: What is the relevance of the stable OVOCs across the lockdown periods in the context of emission sources/anthropogenic activity? Is there any offset between emissions and chemistry during this period?

Response: Thanks for pointing this out. The original description has a mistake. After double check of the data, we found the simulated OVOC concentrations during the three periods were 26.65, 20.75, and 23.80 ppbv, respectively, which follows the same trend as $k_{OH}$. This sentence has been revised to:

As $k_{OH}$ from OVOC, it shared the same trend as OVOC concentration, which reached the minimum value (5.56 s$^{-1}$) during the Full-lockdown period.

Added References:

Adeloye, A. J. and Montaseri, M.: Preliminary streamflow data analyses prior to water resources planning study. Hydrological Sciences Journal 2002; 47, 679–692.

WMO: Analyzing long time series of hydrological data with respect to climate variability, World Meteorological Organization, Geneva, Switzerland 1988.